# Factors Associated with Low Birthweight in Low-and-Middle Income Countries in South Asia

**DOI:** 10.3390/ijerph192114139

**Published:** 2022-10-29

**Authors:** Ngan Ngo, Jahar Bhowmik, Raaj Kishore Biswas

**Affiliations:** 1Department of Health Science and Biostatistics, Swinburne University of Technology, Melbourne, VIC 3150, Australia; 2Charles Perkins Centre, School of Health Sciences, Faculty of Medicine and Health, The University of Sydney, Sydney, NSW 2006, Australia

**Keywords:** low birth weight, low-and-middle income, South Asia, multivariate binary logistic regression, complex survey

## Abstract

Child with Low Birth Weight (LBW) has a higher risk of infant mortality, learning difficulties in childhood due to stunted growth and impaired neurodevelopment, is more likely to develop heart diseases and diabetes in adulthood. This study aimed to evaluate the latest demographic and health surveys (DHSs) across multiple countries in South Asia to determine the factors associated with LBW among these countries. Latest available DHS data across Afghanistan (2015, n = 29,461), Bangladesh (2018, n = 20,127), Nepal (2016, n = 12,862), and Pakistan (2018, n = 15,068) were analysed. Complex survey adjusted generalized linear models were fitted to investigate the association of birth weight with sociodemographic and decision-making factors. Pakistan had the highest proportion of LBW at 18% followed by Afghanistan and Bangladesh at around 14% and Nepal had the lowest (13%). Children born in Pakistan were more likely to have LBW children than Afghanistan (AOR = 2.17, 95% CI = 1.49–3.14). Mothers living in rural areas (AOR = 0.77, 95% CI = 0.61–0.97), with highly educated partners and belonging to richer families were less susceptible to having child with LBW. To reduce 30% LBW in-line with the World Health Organisation’s 2025 goal, policymakers in SA should focus on women in urban areas with low-educated partners belonging to poor households to ease LBW burden.

## 1. Introduction

Child with low birth weight (LBW) have a higher risk of infant mortality, learning difficulties in childhood due to stunted growth and impaired neurodevelopment, and are more likely to develop heart disease and diabetes in adulthood [1,2,3,4]. According to the World Health Organisation (WHO) child born under the weight of 2500 gm is considered LBW. Given the complications associated with it, the WHO has set a goal to reduce cases of LBW by 30% by 2025 [3]. The region of South Asia (SA) currently has the highest proportion (28%) of children who were born with LBW worldwide [3]. It is, thus, deserve careful attention if the 2025 goal is to be achieved.

Several research determined a few factors that were found to be associated with LBW including socio-economic status, maternal age, parity, pregnancy interval, non-pregnant weight, maternal height, haemoglobin level, BMI, trimester bleeding, tobacco consumption, alcohol consumption, gestation age, antenatal care (ANC), and maternal nutritional status [5,6,7]. Furthermore, recent studies demonstrated that women who participates in family decision making was a positive predictor for ANC [8], and women who experienced intimate partner violence (IPV) had a higher risk of giving birth to child with LBW [9].

LBW is one of the major public health concerns, as it is estimated that 20 million children born each year are underweight and is more prevalent in low-middle income countries (LMICs), with 28% of all LBW children born in SA, 13% in sub-Saharan Africa (SSA) and 9% in Latin America (LA) [3]. Recent studies into infant mortality estimated that 60–68% children with LBW worldwide died within 28 days of [10], and a trend analysis between 2013–2018 found that on average birthweight decreased significantly in Africa (mean change = 36.51 g) and Central America (mean change = 53.07 g) but not Asia (mean change = 3.86) [11], hence LBW is trending up in both Africa and Central America but remains constant for Asia which is consistently high. In a recent study, Sathi et al. [12] demonstrated that the prevalence of LBW reported in Afghanistan, Bangladesh, Nepal and Pakistan were 15.13%, 14.93%, 11.73% and 19.18%, respectively, and these were relatively high as compared to other LMICs in the region. The study also found higher level of inequalities among the families regarding their socio-economic status (SES) and wealth index in these countries.

Prior studies into LBW in SA were mostly country specific. This study aimed to address the gap in identifying the common socio-economic factors of LBW in the selected multiple countries to aid policymakers in comparing their policies and assess the action plan or interventions that have worked thus far in achieving target set by the WHO.

This study, thus, aimed to evaluate the latest population surveys of multiple LMICs in SA including Afghanistan, Bangladesh, Nepal, and Pakistan to determine the effects of socio-economic factors, decision-making power of women (DPW), and intimate partner violence (IPV) have on LBW among these countries.

Furthermore, insights into DPW and IPV can setting policies to improve women’s quality of life and to enhance United National gender equality target related to Sustainable Development Goal 5 [13] as well as a step toward empowering women.

## 2. Methods

This study analysed comparable population-based cross-sectional demographic and health surveys (DHS) data across four LMICs in SA conducted between 2015–2021. These surveys collected data on women aged between 15 to 49 years old. The data custodians of DHS ensure the population in surveyed countries are well presented by implementing two-stage stratified sample of households. The first stage involves using probability proportion to size (PPS) to select the most appropriate enumeration areas (EAs) from Census file, the second stage involves using an updated list of households to select the sample of households in each EAs [14].

The most recent DHS survey for Afghanistan (2015) consists of 29,461 women, Bangladesh (2018) with 20,127 women, Nepal (2016) with 12,862 women, and Pakistan (2018) with 15,068 women were used were 98%, 99%, 98% and 96%, respectively [15,16,17]. For the current study, samples where information on child’s weight were available for mother were extracted and the sample size was 2671, 2200, 2564, and 1575 for Afghanistan, Bangladesh, Nepal and Pakistan, respectively. This study focused on latest single birth child for each mother, as few surveyed birth weights were based on mother’s memory and the current demographic information of households are more likely to be comparable with most recent childbirth.

### 2.1. Outcome Variable

The outcome variable for this study was low birthweight—a dichotomous measure with children with birthweight < 2500 gm is considered as ‘LBW’, otherwise noted as ‘normal’ weight.

### 2.2. Covariates

Literature review on LBW in LMICs in SA identified seven covariates as predictors for LBW and was included in this study. These are place of residence (urban/rural), maternal age at birth of child (<20, 20–24, 25–29, and ≥30), respondent’s education level (no education, primary, secondary, and higher), partner’s education level (no education, primary, secondary, and higher), respondent’s employment status (yes/no), wealth index (poorest, poorer, middle, richer, and richest) derived from principal component analysis on data such as household assets owned [18], and respondent’s exposure to media; a dichotomous measure—derived from frequency of watching television, reading the newspaper, and magazine and listening to the radio. For the current analysis, a mother is considered exposed to media if they engaged to any of the media ‘at least once a week’.

Two targeted covariates of this study are DPW and justification to IPV. DPW—a binary variable—depends on respondents’ contribution on three major household decisions such as ‘respondent’s health care’, ‘large household purchases’, and ‘visits to family/relatives’. Respondents were considered to contribute to household decisions if they made these decisions alone or co-decided with partners [8]. Otherwise, if the decision was made by their husband/partner or someone else, they were considered to not have decision-making power. Respondent’s justification to IPV is also a dichotomous variable computed from respondent’s opinion on physical abuse from husband. It is considered ‘yes’ if respondent responded that physically hearting wives was justified if wives did not tell husbands of her whereabouts, neglected children, refused intimacy, and burnt food [9].

### 2.3. Statistical Analysis

This study explored the primary unweighted bivariate association between LBW and the covariates through Chi-square $\left(\chi^{2}\right)$ test. This study assumed data is missing at random and therefore listwise deletion method was used to exclude them from the study, as done in similar studies published previously [19,20]. Survey outcomes of the four countries was merged into one dataset to assess the inter-country differences. To assess the strength and direction of cohort associations the complex-survey generalized linear models (GLMs) adjusted for strata, cluster, and weight variables was fitted on the merged dataset. Adjusted odds ratio (AOR) along with 95% confidence interval was obtained to gain insights on the effect size and strength of association. R package ‘survey’ (*version 4.1-1*) was used to input survey design elements and apply modeling. All analyses were conducted on R (*version 4.1.2*).

## 3. Results

Among the four countries assessed, Pakistan had the highest proportion of LBW at 18.7% and Nepal had the lowest at 11.3%. Both Afghanistan and Bangladesh were similar at around 14%. Pakistan had the highest proportion of LBW in all sociodemographic factors except for mothers with no education/preschool and higher education level. Bangladesh had highest proportion of LBW where mother had no education/preschool, and Afghanistan had highest proportion of LBW where mother had higher education level. Nepal had lowest proportion of LBW in all sociodemographic factors except for material age <20 and richer wealth index, where Afghanistan had lower proportion of LBW.

Results obtained from unweighted bivariate analysis presented in Table 1 show there were slightly higher proportion of LBW children in rural areas than urban areas for all countries except for Afghanistan; however, only strong evidence of difference was found for Pakistan (*p* = 0.011). While the proportion of children with LBW decreases as mother’s age at childbirth increases for Nepal, it is a V-curve for Pakistan and Bangladesh and is inverted-U shape for Afghanistan. Younger mothers (<20-year-old) in Pakistan had much higher proportion of LBW than the other countries, and Afghanistan had the lowest LBW rate (13.9%) compared with Nepal (15.2%) and Bangladesh (15.5%). For mother’s education, while LBW of children in Bangladesh and Nepal decreased as mother’s education level increased, in Pakistan LWB rate was lower among mothers with higher education compared with other education levels. All four countries show the same trend for partner’s education level; LBW decreases when partner’s education level increases that is proportional of children with LBW decreased when partner’s had secondary or higher level.

Mother’s employment status did not show any primary association with LBW of children in the bivariate analysis. Children with LBW were typically more common among mothers residing in lower wealth indices households and wealth index was found to have an association for all for countries. Mothers who were not exposed to media had a higher percentage of children with LBW and showed significant association between LBW and exposure to media for Afghanistan (*p* < 0.001), Nepal (*p* = 0.045) and Pakistan (*p* = 0.001). The proportion of LBW was lower for mothers who contributed to household decisions, and the difference was significant for Nepal (*p* = 0.023) and Pakistan (*p* = 0.046). Women in Afghanistan and Nepal who felt intimate partner violence (IPV) is justified had slightly lower proportion of LBW than those who believed IPV is not justified although the differences are not significant (*p* > 0.05), whereas in Pakistan it is the reversed and the difference is significant (*p* < 0.001).

Table 2 shows the outcome of the survey adjusted binary logistic regression models. Five covariates showed strong evidence of association (*p* < 0.05) with LBW Children from Pakistan had 2.17 times higher odds of being LBW compared to those from Afghanistan (AOR = 2.17, 95% CI = 1.49–3.14). Children were less likely to be born with LBW when mothers live in rural area (AOR = 0.77, 95% CI = 0.61–0.97), have partner with higher education level (AOR = 0.63, 95% CI = 0.42–0.94), and belong to poorer (AOR = 0.71, 95% CI = 0.52–0.97) or richer (AOR = 0.70, 95% CI = 0.50–0.97) wealth quantile compared to poorest households.

## 4. Discussion

This study investigated the factors associated with LBW among children in Afghanistan, Bangladesh, Nepal, and Pakistan using comparable population-based cross-sectional DHS conducted between 2015–2021. The study results showed that the proportion of LBW was lower than previously reported national level studies in the region, with LBW proportions in Afghanistan, Bangladesh, Nepal and Pakistan was 16%, 20%, 15.4% and 25%, respectively and now they are 14.6%, 14.3%, 11.3% and 18.7%, respectively [10,21,22,23].

Among the studied countries, Pakistan needs a greater effort to reduce its higher prevalence of LBW among newborns. This could be because currently Pakistan is behind the other countries in providing adequate medical care for expecting mothers and newborn children [24], and majority of Pakistanis do not spend enough on nutritious diet resulting in malnutrition that could lead to mothers having LBW children [25]. Cultural stigma could be an important factor, not available in the current study data, for Pakistan as well. Ahmed, Khoja and Tirmizi [26] reported that strict religious norms hinder mothers’ access to antenatal care. Furthermore, male dominance in a factor in societies such as Bangladesh and Pakistan, which may have played a role in care for mothers and seeking appropriate medical services [27,28,29].

Although the results obtained from unadjusted bivariate analysis showed a higher proportion of LBW children in rural compared urban areas, especially in Pakistan, the survey weight adjusted model revealed that mothers from urban areas have higher odds of having LBW children compared to their rural counterparts, which is consistent with findings from previous studies in this region [6,7,12]. However, this finding contradicts with some studies where opposite results were reported [30] or no significant difference was found between rural and urban [23,31]. This issue clearly needs further research taking into account the change environmental health in urban setting.

It was found that as fathers’ education level increased the proportion of children’s LBW decreased. Children with well-educated parents are less likely to be LBW and is consistent with previous studies in this region [1,10,21,22,23], this could attributed to educated fathers are more likely to understand the value of maternal health services and nutrition of mother/child. Educated fathers are more aware of maternal health issues and more likely to seek medical expert’s advice during their wife’s pregnancy and time of child delivery which decreases the likelihood of having LBW children.

Mothers belonging to the poorest quantile generally had the highest proportion of LBW children with exceptions in Nepal, and the model results showed strong evidence that higher wealth groups were less likely to have children with LBW. In accordant with previous studies in the same region [22,23,31] this study also found that the likelihood of having LBW child decreased as wealth index increased. Having more money enable mothers to have access to appropriate nutrition, full antenatal care and, able to deliver babies at hospitals by health professionals [31,32].

The strengths of this study are the use of comparable population-based cross-sectional DHSs across multiple countries LMICs in SA providing a snapshot of the whole region as well as comparing country-wise differences. Some interesting avenues for future research were revealed; particularly, the urban rural unexpected finding that rural areas were performing better with lower likelihood of LBW among children. This calls for further studies across different regional areas. Furthermore, education of mother’s, which is typically an indicator for health outcomes in newborns, did not show any strong evidence of association with LBW among children. This could be due to confounding effect and an impact of change in distribution when multiple survey data were merged. The study had some limitations. First, this study used cross-sectional study therefore cannot make causal inferences. Second, this study could not include all LMICs in SA due to lack of available DHS data. Third, as surveys in all four countries were not conducted in the same year there could be minor inconsistencies. Fourth, some confounders such as maternal comorbidity and parity of mothers were not adjusted in the current study models as they were not available in the DHS data sets. Future studies with an extensive data set can explore this. Lastly, the study only included the latest single birth from each household, and some of surveyed birth weights were based on mothers’ memory and can induce some memory bias.

## 5. Conclusions

LBW is a global health issue which is more prevalent in LMICs in SA. To meet the WHO’s target countries in this region can accelerate their relevant public health programs and social policies by leveraging and learn of each other, such as District Investment Case program implemented by Nepal [33] and Inclusive Growth framework implemented by Bangladesh [34]. By evaluating the common factors associated with LBW in the selected LMICs in SA, policymakers are encouraged to focus their attention to the vulnerable group of women living in urban areas with low-educated partners belonging to poorest wealth quantile to decrease the rate of LBW among children in SA.

## Figures and Tables

**Table 1 ijerph-19-14139-t001:** Distribution of sociodemographic factors and proportion of children with LBW in Afghanistan, Bangladesh, Nepal and Pakistan.

Determinants	Afghanistan		Bangladesh		Nepal		Pakistan	
	All (n)	LBW (n/%)	All (n)	LBW (n/%)	All (n)	LBW (n/%)	All (n)	LBW (n/%)
Total	2671	389 (14.6)	2200	317 (14.4)	2564	289 (11.3)	1575	294 (18.7)
Residence								
Urban	1052	154 (14.6)	970	132 (13.6)	1701	180 (10.6)	973	162 (16.6)
Rural	1619	235 (14.5)	1250	185 (14.8)	863	109 (12.6)	602	132 (21.9)
*p*-value	0.974		0.462		0.138		0.011	
Maternal age at birth								
<20	316	41 (13.0)	624	97 (15.5)	578	88 (15.2)	76	25 (32.9)
20–24	842	130 (15.4)	756	105 (13.9)	1038	112 (10.8)	371	79 (21.3)
25–29	666	103 (15.5)	529	65 (12.3)	603	65 (10.8)	549	87 (15.8)
>=30	847	115 (13.6)	311	50 (16.1)	345	24 (7.0)	579	103 (17.8)
*p*-value	0.521		0.328		0.001		0.002	
Mother’s Education								
No Education/preschool	1854	285 (15.4)	57	14 (24.6)	536	74 (13.8)	256	57 (22.3)
Primary	316	37 (11.7)	344	60 (17.4)	435	50 (11.5)	184	41 (22.3)
Secondary	379	52 (13.7)	1126	168 (14.9)	1045	116 (11.1)	504	121 (24.0)
Higher	122	15 (12.3)	693	75 (10.8)	548	49 (08.9)	631	75 (11.9)
*p*-value	0.288		0.002		0.901		<0.001	
Partner’s Education								
No Education, preschool	970	156 (16.1)	158	26 (16.5)	219	35 (16.0)	162	38 (23.5)
Primary	461	74 (16.1)	534	90(16.9)	437	62 (14.2)	150	37 (24.7)
Secondary	847	115 (13.6)	790	127 (16.1)	1285	136 (10.6)	600	121 (20.2)
Higher	393	44 (11.2)	738	74 (10.0)	623	56 (09.0)	663	98 (14.8)
*p*-value	0.076		< 0.001		0.006		0.004	
Employment Status								
Not Working	2418	359 (14.8)	1505	214 (14.2)	1245	139 (11.2)	1363	249 (18.3)
Working	253	30 (11.9)	715	103 (14.4)	1319	150 (11.4)	212	45 (21.2)
*p*-value	0.235		0.958		0.917		0.351	
Wealth Index								
Poorest	299	77 (25.8)	240	47 (19.6)	462	56 (12.1)	77	25 (32.5)
Poorer	405	73 (18.0)	307	39 (12.7)	484	54 (11.2)	184	38 (20.7)
Middle	469	63 (13.4)	399	71 (17.8)	534	74 (13.9)	266	61 (22.9)
Richer	685	82 (12.0)	520	74 (14.2)	572	69 (12.1)	367	72 (19.6)
Richest	813	94 (11.6)	754	86 (11.4)	482	36 (07.5)	681	98 (14.4)
*p*-value	<0.001		0.004		0.048		<0.001	
Exposed to Media								
No	861	159 (18.5)	690	109 (15.8)	949	123 (13.0)	405	99 (24.4)
Yes	1810	230 (12.7)	1530	208 (13.6)	1615	166 (10.3)	1170	195 (16.7)
*p*-value	<0.001		0.190		0.045		0.001	
Decision-making Power of Women								
No	774	106 (13.7)	265	38 (14.3)	720	98 (13.6)	392	87 (22.2)
Yes	1897	283(14.9)	1995	279 (14.0)	1844	191 (10.4)	1183	202 (17.1)
*p*-value	0.452		1.000		0.023		0.046	
Intimate Partner Violence								
No	531	88 (16.6)	1892	266 (14.1)	1867	215 (11.5)	1183	195 (16.5)
Yes	2140	301 (14.1)	328	51 (15.5)	697	74 (10.6)	392	99 (25.3)
*p*-value	0.162		0.531		0.569		<0.001	

**Table 2 ijerph-19-14139-t002:** Results of the logistic regression model fitted to LBW with sociodemographic factors, adjusting for survey weights.

Variable	AOR (95% CI)	*p*-Value
Country (Ref: Afghanistan)		
Bangladesh	1.27 (0.91 to 1.78)	0.159
Nepal	0.73 (0.50 to 1.05)	0.092
Pakistan	2.17 (1.49 to 3.14)	<0.001
Residence (Ref: Urban)		
Rural	0.77 (0.61 to 0.97)	0.027
Maternal age at birth (Ref: < 20)		
20–24	0.91 (0.73 to 1.13)	0.386
25–29	0.86 (0.64 to 1.16)	0.314
>=30	0.76 (0.56 to 1.03)	0.076
Mother’s Education (Ref: No Education/preschool)		
Primary	0.78 (0.58 to 1.05)	0.098
Secondary	0.81 (0.53 to 1.25)	0.334
Higher	0.63 (0.38 to 1.05)	0.079
Partner’s Education (Ref: Reference: No Education, preschool)		
Primary	0.90 (0.66 to 1.23)	0.524
Secondary	0.94 (0.69 to 1.29)	0.707
Higher	0.63 (0.42 to 0.94)	0.024
Employment Status (Ref: Not Working)		
Working	1.07 (0.87 to 1.32)	0.519
Wealth Index (Ref: Poorest)		
Poorer	0.71 (0.52 to 0.97)	0.032
Middle	0.88 (0.66 to 1.20)	0.425
Richer	0.70 (0.50 to 0.97)	0.032
Richest	0.78 (0.52 to 1.19)	0.500
Exposed to Media (Ref: No)		
Yes	0.87 (0.72 to 1.05)	0.143
Decision-Making Power of Women (Ref: No)		
Yes	0.82 (0.66 to 1.03)	0.090
Intimate Partner Violence (Ref: No)		
Yes	1.08 (0.88 to 1.32)	0.467

## Data Availability

This article does not contain any studies with human participants performed by any of the authors. Data used in research was attained from the National Institute of Population Research and Training (NIPORT), and ICF 2020, funded by the United States Agency for International Development (USAID). All identification of the respondents was deidentified before publishing the data. Views expressed in this study do not necessarily reflect those of USAID, the US government, NIPORT, or data custodians. The secondary datasets of the current study are available at https://dhsprogram.com/methodology/survey/survey-display-536 (accessed on 1 May 2022 ). Permission for this project was taken from the Demographic and Health Surveys (DHS) Program authority by the authors.

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
