# Peer review of "Factors Associated with Low Birthweight in Low-and-Middle Income Countries in South Asia"

_ijerph, 2022, doi:10.3390/ijerph192114139_

Round 1

Reviewer 1 Report

This paper investigated the association between demographic characteristics and low birth weight in low- and middle-income countries in South Asia. The work could be improved by more rigorous presentations of the results, as well as some explanations and clarifications:

Abstract:

-OR and CI need to be cited for urban/rural.

Introduction:

-Since the analysis was not able to include all LMICs in SA due to data not being available, it would be helpful if the authors could provide some background information about those four countries in terms of demographic characteristics such as income, LBW rate, SES, compared with other LMICs in SA such that readers could get some better ideas on how results from this study could be extrapolated to other LMICs in SA.

Methods:

-Many important maternal clinical and biometric covariates were not included such as maternal BMI, maternal comorbidity, parity. Therefore, the estimated associations could potentially be confounded by many factors.

-It is important to mention how strata, cluster, and weight variables were adjusted in the regression model.

Results:

-The authors mentioned “Nepal had lowest proportion of LBW in all sociodemographic factors.” This is inconsistent with numbers in Table 1 for maternal age < 20 (15.2% in Nepal vs. 13.0% in Afghanistan) and Wealth Index being richer (12.1 in Nepal vs. 12.0 in Afghanistan)

-“the proportion of children with LBW decrease as mother’s age at childbirth increase for Afghanistan and Nepal” This is not true for Afhganistan, where LBW rate increased then decreased with a V shape.

- “Younger mothers (< 20-year-old) in Pakistan (p = 0.002) had much higher proportion of LBW than the other countries” The authors cited a P-value here, however, the chi-square test compares the distribution of LBW among different maternal age groups in Pakistan. It does not compare any specific age group in Pakistan with other countries. It is incorrect to cite the P-value and the result of chi-square test here.

-“Nepal had lowest proportion of LBW for all age groups (p = 0.001).” Again, this chi-square test only tests whether there is any significant difference between the four age groups in Nepal, not comparing Nepal with other three countries. It is not correct to cite the P-value here.

-“In Pakistan LBW was only low for mothers with Higher education level” It is unclear what the authors were comparing with. Being “low”, the proportion of LBW in the “richer maternal education” group in Pakistan was still higher than other three countries.

- “Mother’s education level for Bangladesh (p=0.002) and Pakistan (p < 0.001).” Incomplete sentence. Not sure what the authors were trying to show.

- “The proportion of LBW is lower for mothers who contributed to household decisions … and similar for those who feel intimate partner violence is justified…” However, based on the observed DPW and IPV distributions in Table 1 the proportion of LBW was similar between yes/no groups in some countries which is inconsistent with the summary description.

- “Children from Pakistan were twice as likely to have LBW than those from Afghanistan (AOR = 2.17, 95%CI = 1.49-3.14).” Incorrect interpretation of OR. OR=2.17 does not mean that the risk of having the outcome was twice than the reference group.

- “…Age at childbirth ≥ 30 years old (AOR = 0.71, 95% CI = 0.56-1.03),” First of all, since maternal age has been shown to have a U (or V) shaped association with LBW based on previous studies, it is recommended to change the reference group where category with the lowest risk (usually 18-35 or 20-35) is used; Second, it is unclear why the authors only selected the category of >= 30 to present while the 20-24 group was also a significant negative association with LBW in this analysis.

-Figure 1 is redundant. All information has already been shown in Table 1.

Discussion:

-It is interesting to discuss about possible reasons that could have driven the distribution of LBW in those countries, for example, in addition to SES could it be possible that some of the cultural factors such as traditional custom also impact peoples’ perspectives; if income could be improved how much of LBW could have been prevented.

-“ The study results showed that the proportion of LBW was lower than previously reported national level studies in the region”. It will be great if authors could show how LBW rate used to look like for those four countries such that readers can compare.

-“Mothers with poorest wealth index have highest proportion of LBW children than any other wealth groups for all countries except for Nepal where it is the middle wealth quantile, and this difference was found to be significant for all countries”. It would be helpful if the authors could mention that this interpretation was based on the chi-square test instead of multivariable regression to avoid any confusions. Also the chi-square test does not compare which category has the highest proportion of LBW. Based on the observed distributions it does not show that the “poorest” group was statistically significantly different from other groups.

-Maternal education was not a significant predictor in the current analysis. However, it does not mean that it was not associated with LBW. It could be impacted by many factors such as distribution, missing status, correlation with paternal education.

-The reversed urban/rural association could also be confounded by other factors on both patient level and country level. More detailed research is needed.

Author Response

Response to reviewer's feedback attched.

Reviewer 2 Report

Introduction

“Child with low birth weight (LBW) have a higher risk of infant mortality, learning 30 difficulties in childhood due to stunted growth and impaired neurodevelopment, and are 31 more likely to develop heart disease and diabetes in adulthood (Lugli et al., 2020; WHO, 32 2014; Zeitlin et al., 2013). According to the World Health Organisation (WHO) child born 33 under the weight of 2500 gm is considered LBW. Given the complications associated to it, 34 the WHO has set a goal to reduce cases of LBW by 30% by 2025 (WHO, 2014). The region 35 of South Asia (SA) currently has the highest proportion (28%) of children who were born 36 with LBW worldwide (WHO, 2014). It is, thus, deserve careful attention if the 2025 goal is 37 to be achieved.”

“28% of all LBW children born in SA, 13% in sub-Saharan Africa (SSA) 50 and 9% in Latin America (LA) (WHO, 2014). Recent studies into infant mortality estimated 51 that 60-68% children with LBW worldwide died within 28 days of birth (M. M. A. Khan 52 et al., 2020), and a trend analysis between 2013-2018 found that on average birthweight 53 decreased significantly in SSA and LA but not SA (Marete et al., 2020), hence LBW is 54 trending up in both SSA and LA but remains constant for SA which is consistently high.”

[ could you please provide quantitative data for these important statements to inform your readers?]

Methods

“This study analysed comparable population-based cross-sectional demographic and 68 health surveys (DHS) data across four LMICs in SA conducted between 2015-2021. These 69 surveys collected data on women aged between 15 to 49 years old. The data custodians of 70 DHS ensure the population in surveyed countries are well presented by implementing 71 two-stage stratified sample of households. The first stage involves using probability pro- 72 portion to size (PPS) to select the most appropriate enumeration areas (EAs) from Census 73 file, the second stage involves using an updated list of households to select the sample of 74 households in each EAs (International, 2012).”

[This is crucial to your study. Could you please provide data on size of the populations studied, the sampling methods and the refusal rate or unable to contact when the surveyors called].  

“This study assumed data is missing at random and therefore listwise deletion method was used to exclude them from the study, as 112 done in similar studies published previously (Bhowmik et al., 2020; Outhwaite & Turner, 113 2007).”

[Which individuals selected for sampling and which data were missing for the individuals sampleD. What are the data that support your statement that missingness is random?]

Results

“LBW of children in Bangladesh and Nepal decreased as mother’s edu- 137 cation level increased, whereas in Pakistan LBW was only low for mothers with Higher 138 education level. Mother’s education level for Bangladesh (p=0.002) and Pakistan (p < 139 0.001). All four countries show the same trend for partner’s education level; LBW de- 140 creases when partner’s education level increases that is proportional of children with LBW 141 decreased when partner’s had secondary or higher level.”

[Please provide a statement in the text of the number of years of education that correlated with each decrement in LBW. This is important for educational planning].

[Please comment on how exposure to media, decision making power of women and intimate partner violence were measured and how different exposures relate to LBW]  

Author Response

(The authors gave the same response as above.)

Round 2

Reviewer 1 Report

The authors addressed most of the comments in a very efficient and appropriate way. There are a couple of points in the results section that should be further improved:

-The authors revised Younger mothers (<20-year-old) in Pakistan had much higher proportion of LBW than the other countries, and Nepal had lowest proportion of LBW for all age groups. The second half of the sentence is inconsistent with numbers in Table 1, where for < 20 group Afghanistan had the lowest LBW rate (13.0%) compared with Nepal (15.2%) and Bangladesh (15.5%).

-The authors revised “whereas in Pakistan LBW was almost 10% lower if mother was highly educated compared to no, primary or secondary education”. This sounds confusing since it is unclear what “10% lower” means. For example, does this mean 22.3% minus 11.9%?  I think it’s more straightforward to say something like “in Pakistan LWB rate was lower among mothers with higher education compared with other education levels”.

-The authors revised “In all countries except for Afghanistan, the proportion of LBW was lower for mothers who contributed to household decisions, and the difference was significant for Nepal (p = 0.023) and Pakistan (p = 0.046)”. For Bangladesh, LBW rate was also similar between yes/no groups, i.e. 14.2% vs. 14.4%. “In all countries except for Afghanistan” needs to be removed.

-The authors revised “Women in Afghanistan and Nepal who felt intimate partner violence (IPV) is justified had lower proportion of LBW than those who believed IPV is not justified”. Since LBW rate for IPV yes/no groups was 14.1% vs. 16.6% in Afghanistan and 10.6% vs.11.5% in Nepal, neither of these comparisons was significant, therefore it should not be concluded that the proportions were different from each other. In my opinion only the difference in Pakistan needs to be cited here.

Author Response

Response to reviewer's feedback attached.

Reviewer 2 Report

Thanks to the authors for their detailed responses to the reviewer and changes to the manuscript.

Author Response

Thank you very much for reviewing our manuscript and providing very useful feedback which has helped us in improving the quality of the manuscript.